# The Best Instruction-Tuning Data are Those That Fit

**Dylan Zhang**
University of Illinois Urbana–Champaign
shizhuo2@illinois.edu

**Qirun Dai**
University of Chicago
qirundai@uchicago.edu

**Hao Peng**
University of Illinois Urbana–Champaign
haopeng@illinois.edu

## Abstract

High-quality supervised finetuning (SFT) data are essential for unlocking pretrained LLMs' capabilities. Typically, instructions are paired with responses from various sources by humans annotators or other LMs, which are often out of the distribution of the target model to be finetuned. This, at scale, can lead to diminishing returns and even hurt the models' performance and robustness. We hypothesize that SFT is most effective with data aligned to the model's pretrained distribution and propose GRAPE— a novel SFT framework that tailors supervision to the target model. For each instruction, it **g**athers **r**esponses from various sources, and selects the one that **a**ligns most closely to the target model's **pre**trained distribution, as measured by the normalized probability. We then proceed with standard SFT with these selected responses. We first evaluate GRAPE with a controlled experiment, where we sample various solutions for each question in UltraInteract from multiple models and finetune on GRAPE-selected data using LMs from different families including LLaMA.1-8B, Mistral-7B and Qwen2.5-7B. GRAPE significantly outperforms strong baselines, including distilling from the strongest model with absolute gain up to 13.8% averaging across benchmarks, and a baseline trained on $3\times$ more data with maximum 17.3% performance improvements. GRAPE's strong performance generalizes to off-the-shelf SFT data. We use GRAPE to subsample responses from the post-training data used for Tulu3 and Olmo-2. GRAPE can outperform strong baselines with 4.5 times the data by 6.1% and state-of-the-art data selection approaches by 3.9% on average performance. Remarkably, using 1/3 data and half number of epochs, GRAPE allows LLaMA.1-8B to surpass the performance of Tulu3-SFT by 3.5%. Our findings highlight that aligning supervision with the pretrained distribution offers a simple yet powerful way to improve SFT efficiency and performance.

## 1 Introduction

High-quality, large-scale supervised data is crucial for supervised fine-tuing(SFT; Databricks, 2023; Köpf et al., 2023; Zhao et al., 2024a; Zheng et al., 2024). A common practice of collecting SFT data involves sampling responses from strong language models, predominantly focusing on expanding the size of the dataset and improving the overall quality of the responses (Sun et al., 2023; Taori et al., 2023; Wang et al., 2023; Xu et al., 2024c; Chen et al., 2024). However, recent research suggests that there is more complex dynamics involved (Xu et al., 2024d). A plateau effect in synthetic data scaling, where performance either stagnates or even declines as the size of the synthetic data increases beyond a certain point, has been widely observed. This phenomenon arises due to issues such as diminishing diversity (Padmakumar & He, 2024; Guo et al., 2023) and distortion in the data

39th Conference on Neural Information Processing Systems (NeurIPS 2025).

distribution (LeBrun et al., 2021), which ultimately undermine the base model's performance and robustness (Alemohammad et al., 2024; Gerstgrasser et al., 2024; Shumailov et al., 2023; Dohmatob et al., 2024; Hataya et al., 2023; Martínez et al., 2023a,b; Bohacek & Farid, 2023; Briesch et al., 2023).

Thus, effective SFT requires more than scaling up the data; it often needs "tailoring" the data to the unique characteristics of the target model. Existing works focus on enhancing the model's existing knowledge and capabilities (Du et al., 2023) and optimizing the curriculum progression for instruction tuning (Zhao et al., 2024b; Lee et al., 2024; Feng et al., 2023; Setlur et al., 2024). They typically find questions the model should best learn, instead of what answers the model should imitate.

Meanwhile, tailoring the responses to the target model has been a crucial ingredient for the success of later phases of LLM development, particularly through on-policy preference learning (Tajwar et al., 2024; Zhang et al., 2024c,a; Miao et al., 2024; Gulcehre et al., 2023; Azar et al., 2023; Tang et al., 2024a; Zhuang et al., 2023), and on-policy/online reinforcement learning (Guo et al., 2024b; Liu et al., 2024d; Zhou et al., 2024c).

Inspired by these insights, we hypothesize that SFT can similarly benefit from aligning data with the model, the core idea behind GRAPE. For each instruction, GRAPE gathers and selects response(s) from various sources that are closest to the target model's pretrained distribution. This is achieved by calculating the probability of each response using the target model and selects the one with the highest length-normalized probability(§3).After obtaining these more "in-distribution" responses, GRAPE proceeds with standard SFT without any modification to the training.

Unlike existing datasets that usually contains one-size-fits-all responses for each instruction without customization (Yu et al., 2024; Yuan et al., 2024b; Lian et al., 2023b; Teknium, 2023), GRAPE curates model-dependent SFT datasets that better matches the base model's distribution, better mitigating the risks associated with distribution shift like spurious correlations (Zhou et al., 2024d) and catastrophic forgetting (Luo et al., 2025; Kotha et al., 2024), while posing minimum overhead of single forward pass over the candidate set. In return, GRAPE allows better downstream performance with reduced training compute.

To GRAPE's advantage, many existing datasets share overlapping instructions but contain different high-quality responses, e.g., the instructions in Flan (Longpre et al., 2023), GSM-8K (Cobbe et al., 2021), MATH (Hendrycks et al., 2021b), and the post-training recipes that re-use SFT instructions for preference learning (Lambert et al., 2024; OLMo et al., 2025). Therefore, GRAPE can directly select, for each model, the best fit(s) among the off-the-shelf responses without having to produce new responses, which proves effective in our experiments (§ 5).

We first validate our hypothesis through extensive controlled experiments on a reasoning dataset with chain-of-thoughts, UltraInteract-SFT (Yuan et al., 2024b), and demonstrate the importance of supervising base models with in-distribution responses(§4). We experimented on 4 popular pretrained LMs from Mistral (MistralAI, 2024c), Llama3.1 (Dubey et al., 2024) and Qwen2.5 (Hui et al., 2024) families. Notably, models fine-tuned with GRAPE-selected responses outperform those trained on a $3\times$ larger datasets up to $17.3\%$ absolute gain on average performances even with less compute, even surpass models trained on responses from the strongest teacher under consideration—LLAMA3.1-405B-INSTRUCT—by significant margins.

We then experiment with a more realistic setting for general-domain instruction-tuning, collecting responses from post-training data for Tulu3 and Olmo-v2. Again, GRAPE demonstrates its effectiveness by outperforming state-of-the-art data selection approaches by avg. $4.6\%$ and a strong baseline trained on all these available data that is 4.5 times larger by up to $6.1\%$. Remarkably, GRAPE allows finetuning a Llama3.1-8B base model to exceed the performance of Tulu-8B-SFT using 1/3 data and half of the epochs.

Our results reveal that distributional alignment is a crucial and previously underappreciated dimension of effective instruction tuning. GRAPE introduces this perspective with a simple, scalable algorithm that consistently outperforms strong baselines with far less data and compute for the actual training.

## 2 Background and Motivation

### 2.1 Data Engineering for Instruction Tuning

Data is central to the success of effective instruction tuning, (Xu et al., 2023; Xia et al., 2024; Chan et al., 2024), featuring both automated data synthesis (Xu et al., 2024a; Zeng et al., 2024; Yu et al., 2024; Wei et al., 2023) and selection (Xia et al., 2024; Chen et al., 2023a; Parkar et al., 2024; Li et al., 2024e). Some selection approaches focus on high-quality data by leveraging LLMs (Chen et al., 2023a; Parkar et al., 2024; Li et al., 2024b) or employing principled metrics (Kang et al., 2024; Mekala et al., 2024; Xia et al., 2024), while others, such as Yang et al. (2024b); Das & Khetan (2023), aim to identify diversity-optimized subsets for greater efficiency. An emerging trend is the customization of training data based on the characteristics of the base models. (Li et al., 2024c; Du et al., 2023; Li et al., 2024a). These methods typically reweight or filter ***instructions***. However, a key overlooked aspect is ***response*** selection—specifically, choosing responses that align with the model's pretrained distribution, which may be critical for preserving useful behaviors and ensuring effective fine-tuning.

### 2.2 Toward Distribution-Aligned Supervised Fine-Tuning

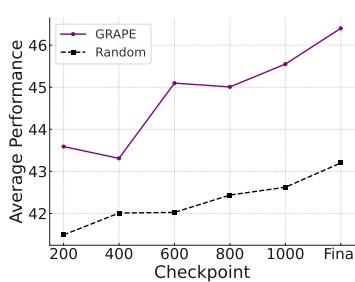

Figure 1: Average performance curve of LLAMA3.1-8B on TULU-3-OLMO-2 combo of GRAPE against random selection

**An Analogy from Reinforcement Learning and Preference Learning** The investigation of this work into the distribution match between the pre-trained LM and supervised fine-tuning (SFT) data is inspired by recent findings on policy optimization for LM alignment with RL (Ouyang et al., 2022) and preference learning (Rafailov et al., 2023; Ethayarajh et al., 2024). While the importance of matching training data distribution with the policy has been well noted in both traditional RL (Shi et al., 2023; Fujimoto et al., 2018; Kumar et al., 2019; Peng et al., 2019; Wang et al., 2021; Arora & Goyal, 2023; Jiang & Li, 2016; Tang & Abbeel, 2010) and LM settings (Xiong et al., 2024), preference learning algorithms like DPO (Rafailov et al., 2023), IPO (Azar et al., 2023) and KTO (Ethayarajh et al., 2024) first emerged as off-policy algorithms. However, subsequent research has highlighted the performance gap between on-policy and off-policy training due to distribution shifts (Xu et al., 2024b; Tang et al., 2024b) and proposed various mitigation strategies (Zhuang et al., 2023; Zhou et al., 2024c; Zhang et al., 2024a; Xiong et al., 2024; Guo et al., 2024b), showing that training models on data more closely aligned with their policy distribution can significantly improve performance, while failing to do so can yield sub-optimal policies or those that are harder to generalize. Works like SPIN (Yuan et al., 2024a) echo the intuition by gradually improving policy through self-play to mitigate distribution shift.

**Hypothesis: Supervised Fine-tuning Benefits From Data That Better Matches Base Distribution** The base distribution of pre-trained language models—shaped by extensive training on vast and diverse datasets—is inherently robust and generalizable (Brown et al., 2020; Saunshi et al., 2021). Therefore, during supervised fine-tuning phase, the pre-trained distribution should be carefully preserved (Kumar et al., 2022; Cohen-Wang et al., 2024; He et al., 2023; Yang et al., 2024d; Ding et al., 2023), to best retain the knowledge and capabilities that emerge during pre-training (Zhou et al., 2023). If the proximity between the pre-trained distribution and the fine-tuning data is not maintained, the limited number of training examples available during SFT, compared to the vast scale of pre-training data, can increase the risk of distribution distortion. This misalignment can lead to issues such as catastrophic forgetting (Aghajanyan et al., 2020; Yang et al., 2024d) and the emergence of spurious correlations (Feldman, 2021).

The central premise of our work is that by using responses closely aligned to the pre-trained distribution, we can minimize distribution shift during SFT and therefore achieve better data efficiency and stronger performance.

**From On-Policy Alignment To Distribution-Aligned SFT** We build on principles of on-policy alignment techniques with key distinctions tailored for SFT. Yet, given that SFT represents an earlier

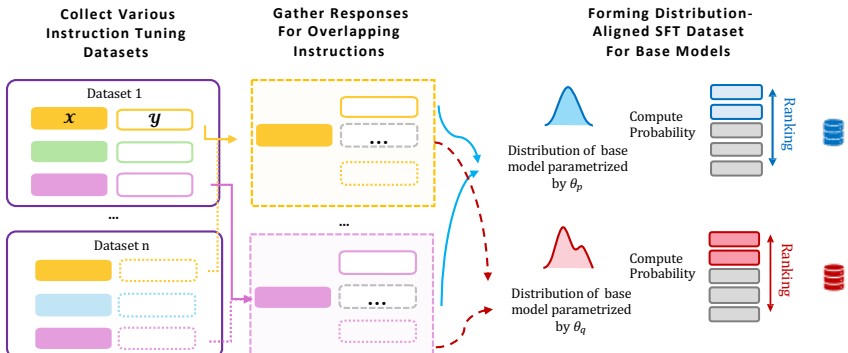

Figure 3: An overview of GRAPE. GRAPE takes multiple off-the-shelf existing datasets and optionally generates new responses, finds overlapping instructions with multiple different responses and selects responses that align with the base model's distribution. This dataset is then used for standard supervised fine-tuning.

stage of post-training than RL, sampling responses from the base model itself alone can lead to model collapse, as noted in (Shumailov et al., 2024). Prior studies have documented similar risks like instability, bias reinforcement, knowledge stagnation, and overfitting (Herel & Mikolov, 2024; Mobahi et al., 2020; Allen-Zhu & Li, 2023; Ghosh et al., 2024; Dong et al., 2025; Zhang et al., 2024d). To address this, we advocate for an approach that stays more in-distribution while delivering effective supervision to the base model. To this end, we propose to gather and select responses from various sources, and select one that is closest to the target model's pretrained distribution, which we name GRAPE.

## 3 Methodology

We introduce GRAPE, a surprisingly simple yet effective methodology to enhance supervised fine-tuning (SFT) by customizing the training data for the base model. The key idea is to find a response, among a candidate pool, for each instruction $x_i$ that aligns closely with the base model's pretrained distribution $\pi_{\theta_0}$.

As diagrammed in Figure 3, GRAPE consists of two main steps, followed by standard SFT:

**Response Collection (§3.1)** Collect a pool of high-quality candidate responses from various sources.

**Customization (§3.2):** For the target model to be fine-tuned $\pi_{\theta_0}$, find the response(s), for each instruction, that are closest to the pretrained distribution of $\pi_{\theta_0}$.

### 3.1 Collecting Responses from Existing Resources

For instruction-tuning of language models, high-quality instructions are more difficult to collect than responses (Xu et al., 2024c; Liu et al., 2024a). Therefore, it is a common practice to reuse existing instruction-tuning prompts while generating diverse responses using various methods tailored to specific requirements. For instance, instructions

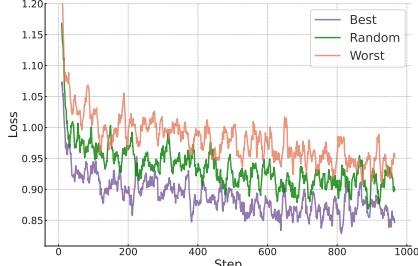

Figure 2: Training loss curve of Llama3.1-8B on TULU-3-OLMO-2. **Best** (highest-probability, selected by GRAPE), **Random**, and **Worst** (lowest-probability) responses show a clear loss ordering: **Best** < **Random** < **Worst**.

from Flan (Longpre et al., 2023), OpenOrca (Lian et al., 2023a), ShareGPT (Team, 2023), and the training splits of GSM-8K (Cobbe et al., 2021), MATH (Hendrycks et al., 2021b), and CodeContests (Li et al., 2022) are frequently reused in datasets like Olmo (OLMo et al., 2025), Tulu (Lambert et al., 2024), OpenHermes (Teknium, 2023), OpenOrca (Lian et al., 2023a), MetaMath (Yu et al., 2024), MathInstruct (Yue et al., 2023), UltraFeedback (Cui et al., 2024), and UltraInteract (Yuan et al., 2024b), whether for SFT or preference learning. The solutions are generated using different

models or follow varying styles depending on the specific needs. This naturally leads to a situation where a single instruction with multiple responses becomes a readily available resource. GRAPE therefore leverages these pre-existing response candidates to tailor training dataset that better aligns with the base model's distribution. When such resources are unavailable or insufficient, practitioners can generate new responses and apply GRAPE.

For each instruction, we collect multiple responses from various datasets. These responses form a candidate set associated with the instruction.

## 3.2 Customize Dataset For Models

We then compute the conditional probability of each response $\pi_{\theta_0}(y_i^j \mid x_i)$. Practically, we format each example using a simple prompt template: `Question: {instruction} \n Answer: {response}`. For each instruction, we rank its candidate responses based on the conditional log-probability assigned by the base model, normalized by response length. This is equivalent to ranking them from lowest to highest perplexity where $\texttt{Perplexity} = \exp\left(-\frac{1}{N}\sum_{t=1}^{N}\log P(x_t \mid x_{<t})\right)$. We then select the responses with the **highest** normalized probability (i.e., lowest perplexity) for supervision. Figure 4 shows a clear difference in models' choices among the same candidate pool, indicating that GRAPE-selected datasets are highly customized towards different models.

Since GRAPE only involves forward-pass log-probability computation (no gradients or optimization), it is highly efficient and simple to integrate into any SFT pipeline with minimal overhead compared with model-based data selection approaches (Xia et al., 2024; Yang et al., 2024b; Liu et al., 2024b; Zhao et al., 2021; Zhang et al., 2024b; Pan et al., 2024). Additionally, it is important to distinguish GRAPE from the other perplexity-based data selection and curriculum planning methods (Wu et al., 2024; Li et al., 2024b; Liu et al., 2024c). Existing approaches focus on selecting **instructions** by using perplexity as a difficulty measure, which differs from GRAPE that uses probability to select for each instruction in a fixed instruction set, responses that better matches with the base model's distribution. Our experiments in §5 demonstrate that low-probability responses with fixed set of instructions are detrimental to performance, further emphasizing the fundamental difference in the two processes.

## 4 Controlled Experiments Show the Benefits of Distributional Alignment

We conduct controlled experiments using UltraInteract-SFT to test whether selecting responses aligned with a base model's distribution improves fine-tuning outcomes more than relying on stronger generators or scaling data size. This setup—focused on verifiable tasks in coding, logic, and math—isolates the effect of distribution matching. Results show that GRAPE-selected responses consistently outperform alternatives, validating distributional alignment as a key supervision signal. These findings motivate our broader evaluations in §5.

### 4.1 Experimental Setup

**Training Data Curation**  In this controlled experiment, we focus on chain-of-thought reasoning (Wei et al., 2022; Wang et al., 2024a; Luo et al., 2024; Cobbe et al., 2021; Li et al., 2023; Lightman et al., 2023). Different models may follow different reasoning paths to solve a problem, while their final solutions can be easily verified.

We use UltraInteract-SFT (Yuan et al., 2024b), which contains approximately $80,800$ unique instructions covering coding, math (chain-of-thought and program-aided) and logic reasoning domains , where each instruction is paired with varying numbers (avg. 3.5/instruction) of different responses to contain a total $> 280,000$ training examples. The responses in the dataset are strictly in step-wise format. For each instruction, we construct a response pool consisting of both (i) original UltraInteract-SFT responses and (ii) additional responses generated from a diverse set of LLMs. GRAPE is then applied to select the most in-distribution response per instruction to this enlarged candidate pool and ensure the number of responses matches the original UltraInteract-SFT dataset for fair comparisons.

We collect responses from a diverse set of models of various sizes across model families, including MIXTRAL-7X7B-INSTRUCT (Jiang et al., 2024), CODESTRAL-22B (MistralAI, 2024a), MISTRAL-SMALL (MistralAI, 2024b), LLAMA-3.1-70B-INSTRUCT and LLAMA-3.1-405B-

INSTRUCT (Dubey et al., 2024), and QWEN2.5-72B-CHAT (Yang et al., 2024a), resulting in approximately 10x additional responses per instruction. The responses are then filtered based on the answers to ensure their validity following Yuan et al. (2024b).

**Base Models** To demonstrate the generalizability of GRAPE, we evaluate its performance across multiple LLMs, including LLAMA-3.1-8B and LLAMA-3.2-3B from LLAMA-3 (Grattafiori et al., 2024) family, MISTRAL-7B (Jiang et al., 2023) and QWEN2.5-7B (Hui et al., 2024). We ensure that all training configurations (GRAPE, baselines) use the same number of instructions and responses per instruction as original UltraInteract-SFT, unless otherwise stated.

**Evaluation** We evaluate the model on coding and math reasoning benchmarks. For coding tasks, we consider HumanEval (Chen et al., 2021), MBPP (Austin et al., 2021), LeetCode (Guo et al., 2024a); for math datasets, we consider **MATH** dataset (Hendrycks et al., 2021b), **GSM-Plus** (Li et al., 2024d) and **TheoremQA** (Chen et al., 2023b) dataset. **HumanEval** and **MBPP** are natural-language-to-code benchmarks testing language models' ability to produce functionally correct programs. **LeetCode** contains interview-level programming problems that are more challenging. **MATH** contains high-school level math competition problems, whereas **GSM-Plus** is a more challenging variant of GSM-8k (Cobbe et al., 2021) and **Theorem-QA** contains complex math reasoning problems.

**Baselines** We compare GRAPE against several baselines to isolate the impact of response selection: **Original Dataset** performs standard SFT on the unmodified UltraInteract-SFT dataset, which includes verified correct responses. **Strongest-Model Responses** uses only responses from the most powerful generator available—LLAMA3.1-405B-INSTRUCT. This represents a strong upper bound on data quality and helps determine whether GRAPE offers additional gains beyond simply choosing a strong model. **3×Data:** use 3 times the number of distinct, validated responses per instruction relative to UltraInteract, while keeping all training hyperparameters (learning rate, number of epochs, etc.) fixed. It directly tests whether GRAPE's gains stem from strategic data selection rather than merely scaling up data volume. [1] We train all models for 1 epoch with a learning rate of $10^{-5}$.

| Model | Data | HE | LC | MBPP | MATH | | GSMPlus | | TheoremQA | | Avg. | Abs. Δ |
| | | | | | CoT | PoT | CoT | PoT | CoT | PoT | | |
|---|---|---|---|---|---|---|---|---|---|---|---|---|
| MISTRAL-7B | Original-UI | 46.3 | 15.6 | 50.1 | 21.6 | 32.6 | 45.9 | 45.3 | 16.8 | 20.1 | 32.7 | *3.6* |
| | Llama3.1-405B | 44.5 | 12.2 | 46.9 | 24.5 | 33.8 | 48.0 | 50.0 | 17.5 | 16.8 | 32.7 | *3.6* |
| | 3x Data | 48.1 | 13.3 | 52.8 | 25.1 | 26.1 | 50.0 | 49.4 | 16.8 | 12.4 | 32.7 | *3.6* |
| | **GRAPE** | **52.4** | **15.6** | **53.4** | **28.9** | **34.6** | **50.5** | **52.8** | **17.8** | **20.6** | **36.3** | - |
| LLAMA3.1-8B | Original-UI | 54.3 | 11.1 | 58.9 | 29.7 | 31.0 | 53.7 | 51.6 | 20.0 | 20.8 | 36.8 | *4.7* |
| | Llama3.1-405B | 56.7 | 15.0 | 60.0 | 34.8 | 38.1 | 51.4 | 55.4 | 16.6 | 21.0 | 38.8 | *2.7* |
| | 3x Data | 48.8 | 7.8 | 57.9 | 25.5 | 11.2 | 48.6 | 45.1 | 20.6 | 19.6 | 31.7 | *9.8* |
| | **GRAPE** | **57.3** | **19.4** | **63.8** | **34.8** | **39.2** | **56.6** | **56.1** | **22.5** | **23.9** | **41.5** | - |
| LLAMA3.2-3B | Original-UI | 32.9 | 3.9 | 41.6 | 12.8 | 16.1 | 30.8 | 19.5 | 14.6 | 10.5 | 20.3 | *3.8* |
| | Llama3.1-405B | 31.7 | 5.0 | 43.3 | 6.6 | 6.6 | 30.8 | 20.6 | 15.1 | 10.8 | 18.9 | *5.1* |
| | 3x Data | 42.6 | 6.7 | 42.9 | 8.7 | 5.1 | 17.8 | 19.5 | 14.6 | **12.8** | 19.0 | *5.1* |
| | **GRAPE** | **42.6** | **13.3** | **44.6** | **16.4** | **17.6** | **34.9** | **20.6** | **15.1** | 11.4 | **24.1** | - |
| QWEN2.5-7B | Original-UI | 67.0 | 41.2 | 60.0 | 51.0 | 38.3 | 64.1 | 59.5 | 14.6 | 10.5 | 45.1 | *11.6* |
| | Llama3.1-405B | 71.3 | 45.0 | 62.0 | 31.5 | 35.1 | 40.1 | 36.4 | 33.3 | 31.2 | 42.9 | *13.8* |
| | 3x Data | 75.6 | 48.3 | 62.9 | 32.8 | 24.5 | 47.1 | 22.8 | 21.6 | 19.0 | 39.4 | *17.3* |
| | **GRAPE** | **77.4** | **48.9** | **70.7** | **56.4** | **45.3** | **67.7** | **66.3** | **37.4** | **40.1** | **56.7** | - |

Table 1: Result of synthetic experiment on UltraInteract-SFT. The last column, **Abs.** Δ is GRAPE's absolute improvement on average performance over that row.

## 4.2 Results and Analysis

Table 1 summarizes the performance of GRAPE across benchmarks. Our approach consistently outperforms the various baselines across the board, including the original UltraInteract-SFT dataset. GRAPE-selected solutions can outperform those directly sampled from the strongest model under consideration (LLAMA3.1-405B-INSTRUCT) up to 13.8% absolute improvement. This implies that customization for base models should be prioritized over identifying the presumably highest-quality responses. This verifies our central premise that being in-distribution with each base model is an important ingredient for the responses we supervise the base models on, to boosting downstream performance. Furthermore, we demonstrate that merely adding more responses does not always lead

---

[1] For instance, if UltraInteract contains 3 validated responses for instruction $x$, this setting uses 9.

to continuous improvement in model performance, which aligns with findings in prior studies (Li et al., 2024c; Du et al., 2023). By properly aligning with models' base distributions, GRAPE outperforms those trained with 3x responses with at least 3.6% and up to 17.3% absolute improvement. These results reinforce the notion that scaling data without considering its alignment with the base model's initial distribution risks diminishing returns and, in some cases, even performance degradation.

## 5  GRAPE-Picking From Real-World SFT Datasets

In this section, we leverage the findings from the earlier experiments and demonstrate the effectiveness of GRAPE to customize training data for each base model by selecting from available datasets with overlapping instructions. Here, we do *not* generate any new responses for the instructions; it only selects from existing ones. We evaluate GRAPE on the fully open dataset used in post-training phases of TULU-3 (Lambert et al., 2024) and OLMO-2 (OLMo et al., 2025). The details are presented below. We discuss additional results in Appendix B

### 5.1  Data Mixture Details

TULU-3 (Lambert et al., 2024) is a fully open-source collection of post-training recipes, including supervised fine-tuning and preference alignment data. OLMO-2 (OLMo et al., 2025) is a fully open-source language model. Both TULU-3 and OLMO-2 use the same data mixture during the supervised fine-tuning stage, but different data mixtures and source models for generating preference data for different sizes of their models: Tulu-3-8B/70B and Olmo-2-7B/13B. To demonstrate the effectiveness of GRAPE, we collected the overlapping instructions from both models and gather their corresponding responses.

From the preference data, we retained only the winning responses. We formed our candidate pool with those instructions with at least two distinct responses, resulting in a dataset of 350.4K unique instructions and about 1.03 million total instruction-response pairs for evaluation with GRAPE. We do *not* apply further processing of these data or any filtering on top of GRAPE.

### 5.2  Evaluation

We evaluate on a set of commonly used benchmarks spanning over coding, math, knowledge and instruction-following. We evaluated on LeetCode (Guo et al., 2024a), MATH (Hendrycks et al., 2021b), Big-BenchHard(BBH) (Suzgun et al., 2022), MMLU (Hendrycks et al., 2021a), and AlpacaEval-V2 (Dubois et al., 2024). Leet-Code, MATH, BBH and MMLU are evaluated the in the same way as in (Yuan et al., 2024b), where we use zero-shot for MATH

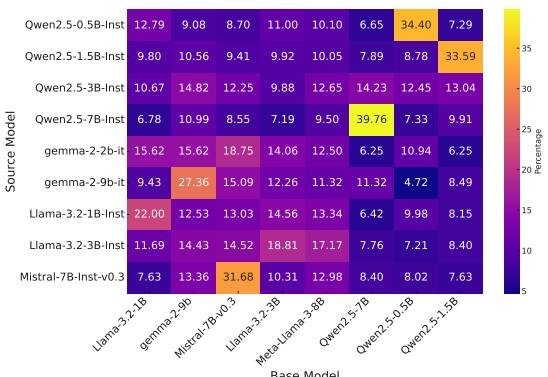

Figure 4: Breakdown of GRAPE-selected responses for 1K Tulu instructions vary significantly across base models, reflecting its highly model-oriented nature over responses. Details in Appendix H

and MMLU, 3-shot example for BBH. We use the same AlpacaEval-v2 as in OpenInstruct.

### 5.3  Baselines

We extensively compare against three baseline types: controlled baselines with fixed instructions, scaling baselines with increased data, and state-of-the-art selection methods. Additional details in Appendix C

**Controlled Baselines**   We include three baselines to isolate the effect of GRAPE's response selection. **SFT-only** replaces GRAPE-selected responses with those from the original SFT dataset, pairing each instruction with a standard reference response to measure improvement over presumably good

SFT responses. **Random** selects candidate responses uniformly at random from the pool for the same set of instructions, establishing a noise-tolerant baseline. **Reverse-GRAPE** instead selects responses with the highest perplexity. We test if responses diverging the most from the base model's distribution degrade performance, providing a contrast that sharpens the effectiveness of GRAPE.

**Scaling Baselines** To assess how GRAPE compares under larger-scale training, we consider three scaling-oriented baselines. **Tulu3-SFT** uses all 939K SFT instances from the Tulu3 training mixture to test whether GRAPE can still outperform despite using only a subset of the instruction pool. **All Responses** trains over the entire 1.04M-instance candidate pool to demonstrate the effect of selection versus brute-force inclusion. **All Available Data** uses all 1.58M instruction-response pairs under consideration, roughly $4.5\times$ the data used by GRAPE, to test whether data volume alone suffices.

**State-Of-The-Art SFT Data Selection Approaches** We compare GRAPE against recent state-of-the-art data selection methods. **LESS** (Xia et al., 2024) selects data based on influence scores on validation tasks. We follow the implementation setup from Dai et al. (Dai et al., 2025). **Emb-NV** selects training data close to validation in embedding space using NV-Embed-V2 (Lee et al., 2025), following (Ivison et al., 2025). **S2L** (Yang et al., 2024b) clusters data via loss trajectories from small reference models (LLAMA-3.2-1B, QWEN2.5-0.5B, MISTRAL-V0.3-7B) and samples uniformly across clusters to match GRAPE's data budget.

| Model | Data | Num. Instances | AlpacaEval2 LC | AlpacaEval2 WR | BBH | MMLU | MATH | LeetCode | Avg. | Abs. Δ |
|---|---|---|---|---|---|---|---|---|---|---|
| LLAMA 3.1-8B | Highest | 350.4k | 10.1 | 6.2 | 68.9 | 63.2 | 22.9 | 13.3 | 30.8 | 5.2 |
| | Random | 350.4k | 12.8 | 10.8 | 68.6 | 63.1 | 27.9 | 13.3 | 32.8 | 3.2 |
| | SFT-Only | 350.4k | 7.1 | 5.5 | 68.9 | 64.1 | 20.2 | 17.2 | 30.5 | 5.4 |
| | Tulu3-SFT | 939k | 12.4 | 8.0 | 67.9 | 65.9 | 31.5* | 7.8 | 32.4 | 3.5 |
| | All Responses | 1.03M | 12.9 | 11.4 | 68.7 | 62.8 | 32.1 | 17.2 | 34.2 | 1.8 |
| | S2L | 350.4k | 8.5 | 7.6 | 68.6 | 63.1 | 26.5 | 16.1 | 31.7 | 4.2 |
| | Emb-NV | 350.4k | 8.4 | 7.0 | 67.9 | 63.7 | 32.1 | 17.2 | 32.7 | 3.2 |
| | LESS | 350.4k | 7.3 | 6.0 | 68.2 | 63.3 | 25.1 | 16.1 | 31.0 | 4.0 |
| | All Available Data | 1.58M | 8.8 | 10.1 | **69.8** | 62.1 | **32.5** | 16.1 | 33.2 | 2.7 |
| | **GRAPE** | 350.4k | **14.8** | **15.2** | 69.6 | **64.5** | 32.1 | **19.4** | **35.9** | - |
| MISTRAL -7B | Highest | 350.4k | 7.0 | 5.5 | 58.8 | 56.1 | 15.1 | 10.6 | 25.5 | 6.4 |
| | Random | 350.4k | 10.6 | 9.2 | 60.0 | 57.8 | 19.6 | 11.1 | 28.1 | 3.9 |
| | SFT-Only | 350.4k | 7.1 | 5.3 | 53.9 | 57.0 | 14.4 | 12.0 | 24.9 | 7.0 |
| | Full-SFT-Data | 939k | 11.5 | 10.5 | 59.0 | 55.2 | **25.8** | 15.6 | 29.9 | 2.0 |
| | All Responses | 1.03M | 10.5 | 11.5 | 61.0 | 57.9 | 24.2 | 14.4 | 29.9 | 2.0 |
| | S2L | 350.4k | 10.5 | 11.9 | 61.9 | 57.1 | 22.4 | 13.9 | 29.6 | 2.3 |
| | Emb-NV | 350.4k | 6.0 | 5.2 | 60.7 | 56.5 | 23.9 | 11.7 | 27.3 | 4.6 |
| | LESS | 350.4k | 6.4 | 4.8 | 59.2 | 55.4 | 16.0 | 8.3 | 25.0 | 6.9 |
| | All Available Data | 1.58M | 8.0 | 7.0 | 55.3 | 53.7 | 25.4 | 12.3 | 26.9 | 5.0 |
| | **GRAPE** | 350.4k | **13.6** | **13.9** | **62.3** | **59.2** | 24.2 | **18.3** | **31.9** | - |
| QWEN2.5 -7B | Highest | 350.4k | 8.0 | 10.7 | 72.2 | 73.2 | 49.4 | 42.2 | 42.6 | 5.9 |
| | Random | 350.4k | 16.1 | 14.9 | **73.3** | 73.1 | 56.0 | 43.3 | 46.1 | 2.4 |
| | SFT-Only | 350.4k | 9.9 | 7.8 | 71.2 | 74.1 | 51.1 | 46.6 | 43.4 | 5.1 |
| | Full-SFT-Data | 939k | 9.5 | 7.1 | 71.4 | 73.1 | 47.0 | **48.3** | 42.7 | 5.8 |
| | All Responses | 1.03M | 16.0 | 14.5 | 71.4 | 72.1 | 51.7 | 43.3 | 44.8 | 3.7 |
| | S2L | 350.4k | 13.4 | 14.9 | 72.7 | 73.1 | 53.4 | 40.6 | 44.7 | 3.9 |
| | Emb-NV | 350.4k | 11.9 | 10.6 | 72.1 | 72.5 | 53.3 | 43.3 | 44.0 | 4.6 |
| | LESS | 350.4k | 7.8 | 5.9 | 71.3 | 72.9 | 48.3 | 41.1 | 41.2 | 7.4 |
| | All Available Data | 1.58M | 13.3 | 12.3 | 70.3 | 71.8 | 44.0 | 42.8 | 42.4 | 6.1 |
| | **GRAPE** | 350.4k | **20.0** | **20.4** | 73.2 | **73.3** | **60.0** | 44.4 | **48.6** | - |

Table 2: GRAPE on the Tulu-Olmo collection. For Llama3.1-8B base model, we included Tulu3-SFT model's results. The "*"-marked number for MATH is obtained using 4-shot prompting. We train all the models (except that we took Tulu3-SFT numbers directly) for 1 epoch with a learning rate of $10^{-5}$. Abs.Δ is GRAPE's absolute improvement on average performance over that row.

## 5.4 Results

As shown in Table 2, models fine-tuned on responses selected by GRAPE outperforms the strong baselines we constructed, especially the one that trains over all available data by significant margins across the 3 models.

| | GRAPE | QWEN2.5 -72B | LLAMA3.1 -405B | GEMMA -IT-9B |
|---|---|---|---|---|
| **LC** | **28.1** | 25.8 | 16.3 | 26.9 |
| **WR** | **33.1** | 24.3 | 17.0 | 20.2 |

Table 3: Alpaca-Eval2 On Magpie-Zoo Xu et al. (2024d).

Remarkably, using roughly 1/6 training compu-
tation (Tulu3-8B-SFT was trained for 2 epochs
on 3 times of data), our performance exceeds that of TULU3-8B-SFT.

Also, GRAPE outperforms state-of-the-art data-selection approaches like S2L, despite its simplicity
and efficiency, further highlighting its effectiveness in diverse real-world scenarios and making it a
practical option for real-world SFT setups with minimal data engineering effort. Without the need
to synthesize any new data, one can easily leverage established datasets sourced from the web to
customize a dataset for each base model that yields better fine-tuning outcome.

These results highlight GRAPE as an effective and efficient selection strategy for real-world SFT.

## 5.5   Ablations

**Remains Effective Even with a Single Response Source**   In Section 4, we demonstrated how
GRAPE enables practitioners to refine model-generated responses for improved training outcomes.
Beyond that, GRAPE can optimize responses from a single generator. The Magpie-Zoo (Xu et al.,
2024d) dataset contains a fixed set of instructions and multiple versions of response sets each
generated by different language models. Using the Magpie-Zoo instruction set, we sample 10
responses per instruction from Qwen2.5-72B-Instruct, select in-distribution responses with GRAPE,
to train a Mistral-v0.3-7B model. We compare with replicas that achieve top-3 performance from
Magpie-Zoo.

As shown in Table 3, GRAPE-selected responses further
boost the performance.  Given that batch sampling from
a strong model introduces minimal latency (Zhong et al.,
2024; Zhou et al., 2024e), this result positions GRAPE as
a practical and efficient data curation strategy—achieving
strong results with just a single generator and no additional
engineering overhead.

| Model | Data | Avg. |
|---|---|---|
| MISTRAL-7B | Self-Distilled | 28.4(-) |
| | Original-UI | 32.7 |
| LLAMA3.1-8B | Self-Distilled | 29.4(-) |
| | Original-UI | 36.8 |
| LLAMA3.2-3B | Self-Distilled | 15.1(-) |
| | Original-UI | 20.3 |

Table 4: Performance Degradation From
Self-Distillation On UI.

**Why GRAPE Outperforms Self-Generated Responses**
To probe the effectiveness of GRAPE, we ablate it against
a degenerate alternative: self-generation, where the model is fine-tuned on its own outputs.

| Dataset | Model | Response Generator | N | Acc. |
|---|---|---|---|---|
| MATH | MISTRAL | Llama3.1-70B | 10 | 18.2 |
| MATH | MISTRAL | FT-Mistral | 10 | 15.9 (-) |
| MATH | LLEMMA | Llama3.1-70B | 10 | 26.2 |
| MATH | LLEMMA | FT-Llemma | 10 | 23.6 (-) |
| MATH | MISTRAL | MM-AnsAug | – | 22.3 |
| MATH | MISTRAL | FT-Mistral | 10 | 20.6 (-) |
| MATH | LLEMMA | MM-AnsAug | – | 28.1 |
| MATH | LLEMMA | FT-Llemma | 10 | 21.4 (-) |

Table 5: Self-generation on MATH dataset. FT-MISTRAL
refers to the model right above that row finetuned from either
MM-AnsAug or Llama3.1-70B-Instruct produced solutions.
N stands for the number of responses sampled.

We fine-tune a base model using re-
sponses generated by its previously
fine-tuned variant.  This setup con-
sistently degrades performance (Ta-
ble 4).  We further confirm this on
the MATH dataset (Hendrycks et al.,
2021b): responses from strong mod-
els like LLAMA3.1-70B-INSTRUCT
or MetaMathQA (Yu et al., 2024) are
used to fine-tune a model that then
generates new solutions for another
round of fine-tuning.  Again, perfor-
mance drops (Table 5).

This failure arises from distributional
collapse (see § 2.2): self-generated re-
sponses become increasingly narrow

and repetitive, reinforcing biases and reducing exposure to diverse reasoning. Correctness alone is
insufficient—external diversity is essential for generalization.

GRAPE avoids this collapse by selecting external responses that are both diverse and distribution-
aligned, preserving semantic breadth and stylistic variability while staying true to the model's
pretraining. This enables more stable and generalizable fine-tuning.

# 6  Conclusion

We present GRAPE, a simple yet effective method for improving supervised fine-tuning by selecting responses aligned with the base model's pretrained distribution. GRAPE requires only a forward pass over candidate responses, making it highly efficient and easy to integrate. Despite its simplicity, GRAPE consistently outperforms stronger baselines using significantly larger datasets and surpasses more complex, costly data selection methods. Our study affirms that carefully aligning SFT data with a model's pretrained distribution yields substantial performance and efficiency gains.

# 7  Discussion and Limitations

**Response versus Instance Level Selection**   GRAPE selects responses: it begins with a fixed set of instructions and evaluates multiple candidate responses for each. This differs from instance-level selection approaches (e.g., Kung et al. (2023); Li et al. (2024c); Wang et al. (2024b)), which focus on choosing which instructions to include based on factors such as coverage, skill-balancing or difficulty. In contrast, GRAPE focuses on the quality of supervision—that is, selecting responses to provide the most effective learning signal. Importantly, this response-centric perspective is complementary, not contradictory, to instance-level or instruction-based selection methods; both might be combined to enhance overall data quality and training efficiency.

**Limitations**   Like many other data selection algorithms (Du et al., 2023; Li et al., 2024c; Xia et al., 2024; Das & Khetan, 2023; Kang et al., 2024; Mekala et al., 2024; Yang et al., 2024c; Zhang et al., 2024b; Pan et al., 2024; Dai et al., 2025; Ivison et al., 2025; Bhatt et al., 2024; Yin & Rush, 2024; Liu et al., 2024b), GRAPE assumes a quality-controlled candidate pool from which to select samples. Furthermore, because GRAPE relies on the base model itself, its selection effectiveness may be influenced by the model's inherent capabilities.

# 8  Acknowledgment

This project is partly supported by NSF under award No. 2019897. This research used the DeltaAI advanced computing and data resource, which is supported by the National Science Foundation (award OAC 2320345) and the State of Illinois. DeltaAI is a joint effort of the University of Illinois Urbana-Champaign and its National Center for Supercomputing Applications.

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

# A  Parameter Distance

We measure the L2 norm of model parameter difference between fine-tuned and pre-trained checkpoints, as a signal of how much the distribution has drifted during SFT ichi Amari (2016); Cover & Thomas (2006). We notice that training over well-matched distribution shifts the parameter less than training over those ill-matched.

|  | **Mistral-7B-v0.3** | **Llama3.1-8B** | **Qwen2.5** |
|---|---|---|---|
| **GRAPE** | 8.006 | 8.196 | 8.426 |
| **Worst** | 8.029 | 8.202 | 8.467 |

Table 6: Performance comparison across different models

# B  Further Experiments

## B.1  Comparing with Reward Based Selection

We compare GRAPE with purely reward-based selection. where for each instruction, we select the response with the highest scalar reward as determined by a reward model - Skywork-Reward-Llama3.1-8B-v0.2. Once the top-ranked response for each instruction is selected, we proceed with standard supervised fine-tuning on the resulting instruction-response pairs.The RFT setup provides a natural contrast to our proposed GRAPE method by emphasizing reward alignment over base-model alignment, thereby enabling us to disentangle the effects of distribution matching versus reward optimization in SFT data selection.

As shown in Table B.1, GRAPE outperforms reward-based selection across both models and all benchmarks. These results suggest that aligning supervision with the base model's own distributional preferences—rather than relying on external reward models—can yield better task performance.

| Model | Method | AE WR | AE WR (LC) | LeetCode | MATH | MMLU | BBH | Avg |
|---|---|---|---|---|---|---|---|---|
| LLaMA3.1-8B | GRAPE | 15.2 | 14.8 | 19.4 | 32.1 | 64.5 | 69.6 | **35.9** |
|  | Reward | 14.0 | 14.5 | 17.2 | 31.3 | 63.3 | 69.0 | 34.9 |
| Mistral-v0.3-7B | GRAPE | 13.9 | 13.6 | 18.3 | 24.2 | 59.2 | 62.3 | **31.9** |
|  | Reward | 12.5 | 13.9 | 13.3 | 22.4 | 58.5 | 62.2 | 30.5 |

Table 7: Performance comparison between GRAPE and reward-based selection across benchmarks for LLaMA3.1-8B and Mistral-v0.3-7B. Metrics are benchmark-specific scores (higher is better).

## B.2  Experiment on OpenHermes

To test the generality of our findings beyond the UltraInteract and Tulu-Olmo settings, we conduct additional experiments on the OPENHERMES-2.5 (Teknium, 2023) dataset—a large-scale, high-quality instruction-tuning corpus with approximately 1 million distinct instructions.

Following the setup from §5, we apply GRAPE to select from responses aggregated across sources, including Huang et al. (2024) and HuggingFace-H4 (2024). For preference-based datasets, we retain only the winning responses to ensure quality, mirroring our earlier selection protocol. This results in 575K unique instructions and 1.34M instruction-response pairs.

| Item | Metric | Data | Llama 3.1-8B | Mistral -7B-v0.3 | Qwen 2.5-7B |
|---|---|---|---|---|---|
| **Alpaca -Eval2** | LC | Subset | 8.6 | 5.9 | 7.6 |
|  |  | Random | 8.0 | 6.2 | 9.0 |
|  |  | **GRAPE** | **11.3** | **8.2** | **10.8** |
|  | WR | Subset | 6.2 | 3.9 | 5.2 |
|  |  | Random | 6.4 | 4.8 | 7.2 |
|  |  | **GRAPE** | **9.4** | **7.5** | **9.6** |
| **Truthful -QA** | MC2 | Subset | 51.6 | 49.0 | 54.4 |
|  |  | Random | 51.4 | 49.9 | 55.2 |
|  |  | **GRAPE** | **52.7** | **51.6** | **56.4** |

Table 8: Results on OpenHermes-2.5. The Subset row refers to training exclusively on the SFT responses over the subset.

As shown in Table 8, GRAPE continues to outperform naive combination strategies. The consistent gains across diverse data sources and model families strengthen our central claim: GRAPE is a general-purpose, model-aligned response selection strategy that reliably improves SFT performance in real-world, large-scale instruction tuning.

## B.3 Data Selection For Long Chain-of-Thoughts

O1-/R1-style long chain-of-thoughts have drawn increasing attention. This paradigm, exemplified by models like OpenAI's O1 and DeepSeek-R1, has shown remarkable success in challenging domains such as mathematics and coding. We further experiment with the use of GRAPE in long chain-of-thought distillation. We generate multiple candidate trajectories using R1-Distill-Qwen-1.5B DeepSeek-AI et al. (2025) for a subset of OpenR1-Math (Face, 2025) dataset — LUFFY (Yan et al., 2025), and verify the correctness of each, retaining the correct ones.

We compare the results on lowest versus highest perplexity instances below in table 9.

| Model | Perplexity | Acc. |
|---|---|---|
| Qwen2.5-1.5B | Highest | 0.330 |
| Qwen2.5-1.5B | Lowest (GRAPE) | 0.396 |
| Qwen2.5-3B | Highest | 0.524 |
| Qwen2.5-3B | Lowest (GRAPE) | 0.544 |

Table 9: Performance metrics on MATH dataset

## B.4 Token-level GRAPE

To further investigate the alternative uses of our insight, we conduct experiments beyond data selection by incorporating token-level likelihoods directly into the training objective. Specifically, we modified the loss function to weigh each token proportionally to its likelihood. We trained this variant on the OpenThoughts-114k Team (2025) dataset, a curated collection of 114k high-quality reasoning samples spanning domains such as math, science, code etc. Evaluation followed the same benchmark suite as LUFFY Yan et al. (2025), including competition-level math datasets (AIME24/25, AMC, MATH-500, Minerva, OlympiadBench) and general reasoning tests (ARC-c, GPQA-Diamond, MMLU-Pro). Our result in Table 10 show that our token-level likelihood-weighted training yields consistent improvements across several benchmarks.

| Benchmark | Baseline (%) | Token-level GRAPE (%) |
|---|---|---|
| MATH | 55.60 | 60.20 |
| OLYMPIADBENCH | 21.04 | 28.30 |
| MINERVA | 15.07 | 19.85 |
| AIME-24 | 3.12 | 4.90 |
| AMC | 23.76 | 29.74 |
| AIME-25 | 4.38 | 3.23 |
| ARC-C | 60.84 | 76.62 |
| GPQA-DIAMOND | 7.07 | 19.70 |
| MMLU-PRO | 27.02 | 34.35 |

Table 10: Qwen2.5-3B performance (in %) on various benchmarks under Baseline vs. Token-level GRAPE.

## C Further Details On Baselines

This section details the experimental setup for our data selection baselines: **S2L**, **LESS** and **NV-Embed**.

### C.1 S2L

S2L, a state-of-the-art unsupervised data selection baseline, operates through two key steps: training a reference model to capture training dynamics and clustering the resulting trajectories to form a

diverse, balanced subset of training data. The reference models used in our setup are specifically selected to enhance S2L's performance, adhering to the theoretical underpinnings from the original paper that training dynamics remain consistent across models of varying sizes within the same family.

For our experiments, we train small reference models corresponding to the final target models. Specifically, we pair Llama-3.1-8B with Llama-3.2-1B, Qwen-2.5-7B with Qwen-2.5-0.5B, and Mistral-v0.3-7B with itself due to the lack of smaller models in the Mistral family. To minimize computational costs, LoRA is applied when training the Mistral reference model. **This choice of reference models are better compared to original S2L setup**, which employed a Pythia-70M proxy, thereby improving the fidelity of the selected subset.

Following S2L, the reference models are trained on a random 5% subset of the dataset over four epochs. This reduced training requirement is justified by prior work, which demonstrates that only partial data is sufficient for the proxy model to learn meaningful training dynamics. During trajectory collection, we record the training loss of all examples at intervals of 500 iterations. The batch size and learning rate schedules are set as batch size of 128 and a learning rate warmup of 3%, followed by a cosine decay to 2e-5.

We then perform K-means clustering using the Faiss library to efficiently partition the trajectory space into 100 clusters. The number of iterations is set to 20, and we use the Euclidean distance metric to ensure convergence to well-separated clusters. From each cluster, an equal number of examples are sampled to maintain a balanced subset distribution.

### C.2 LESS

LESS is a state-of-the-art model-based and supervised data selection method that leverages gradient-based influence estimation. Given a small set of validation examples per task, LESS computes the influence of each training example by measuring the weighted cosine similarity between their LoRA gradients across multiple warmup checkpoints. It then aggregates these influence scores by averaging over validation examples within each task, followed by taking the maximum across tasks to obtain a scalar utility score per training example. Training examples are selected greedily based on these scores. In our experiments, we use the same base model for both selection and training, and follow the original LESS setup: 5% warmup training for 4 epochs and a gradient projection dimension of 8192.

### C.3 NV-Embed

Embedding-based data selection as detailed in (Ivison et al., 2025) is a supervised data selection method that ranks training examples by computing cosine similarity between their embeddings and those of validation examples. Unlike model-aware methods like LESS, embedding-based data selection is model-agnostic: it relies on fixed, pretrained embedding models (in our case, we used NV-embed-v2, the state-of-the-art embedding model) rather than the target model. Instead of aggregating similarity scores into a single utility value per training example, embedding-based data selection uses a round-robin strategy that iteratively selects the highest-scoring example for each validation instance, ensuring diverse coverage across tasks. We follow the original setup from Ivison et al. (2025) in our experiments.

## D Further Training Details

We train our models on a 4-GPU Nvidia-GH200 node, with batch size 256 and micro batch size 2.

## E Further Ablations on UltraInteract.

See Tables 11 and 4.

## F Additional Related Works On Model Dependent Data Selection Approaches

Model-dependent data selection methods leverage internal signals from a target model—such as gradients, embeddings, or log-probabilities—to identify training examples that are most useful for

| Data | | Full UI | Closest-1 | Random-1 |
|---|---|---|---|---|
| Num. Instances | | 280K | 80K | 80K |
| HumanEval | | 46.3 | 42.1 | 41.5 (-) |
| LeetCode | | 15.6 | 13.9 | 11.1 (-) |
| MBPP | | 50.1 | 52.1 | 49.1 (-) |
| MATH | CoT | 21.6 | 19.2 | 15.5 (-) |
| | PoT | 32.6 | 24.9 | 15.1 (-) |
| GSMPlus | CoT | 45.9 | 44.1 | 35.3 (-) |
| | CoT | 45.3 | 43.2 | 45.2 (-) |
| TheoremQA | CoT | 16.8 | 15.8 | 15.8 |
| | PoT | 20.1 | 12.9 | 15.3 |
| Avg. | | 32.7 | 29.8 | 27.1(-) |

Table 11: Ablations on data selection with MISTRAL-7B-V0.3 by selecting within UltraInteract-SFT (since it contains varying numbers of responses per-instruction). Closest-1 denotes the one closest to the base model's initial distribution. Random-1 is sampled from the entire enlarged dataset formed by both original and generated responses. We use (-) to denote **Random-1** underperforming **Closest-1**.

| Model | Data | HE | LC | MBPP | MATH | | GSMPlus | | TheoremQA | | Avg. |
|---|---|---|---|---|---|---|---|---|---|---|---|
| | | | | | CoT | PoT | CoT | PoT | CoT | PoT | |
| Mistral-7B | Self-Distill | 46.3 | 13.3 | 49.6 | 17.3 | 18.5 | 43.3 | 33.2 | 16.8 | 17.4 | 28.4 |
| | Original-UI | 46.3 | 15.6 | 50.1 | 21.6 | 32.6 | 45.9 | 45.3 | 16.8 | 20.1 | 32.7 |
| | Ours | 52.4 | 15.6 | 53.4 | 28.9 | 34.6 | 50.5 | 52.8 | 17.8 | 20.6 | 36.3 |
| Llama3.1-8B | Self-Distilled | 47.6 | 6.7 | 51.7 | 22.9 | 12.7 | 47.2 | 35.3 | 18.8 | 21.5 | 29.4 |
| | Original-UI | 54.3 | 11.1 | 58.9 | 29.7 | 31.0 | 53.7 | 51.6 | 20.0 | 20.8 | 36.8 |
| | Ours | 57.3 | 19.4 | 63.8 | 34.8 | 39.2 | 56.6 | 56.1 | 22.5 | 23.9 | 41.5 |
| Llama3.2-3B | Self-Distilled | 32.3 | 5.6 | 41.9 | 8.8 | 7.0 | 12.1 | 12.1 | 5.9 | 10.5 | 15.1 |
| | Original-UI | 32.9 | 3.9 | 41.6 | 12.8 | 16.1 | 30.8 | 19.5 | 14.6 | 10.5 | 20.3 |
| | Ours | 42.6 | 13.3 | 44.6 | 16.4 | 17.6 | 34.9 | 20.6 | 15.1 | 11.4 | 24.1 |

Table 12: The detailed comparison across benchmarks for self-distillation discussed in Section 5.5

fine-tuning. These approaches have led to strong empirical results across various settings. However, many of them involve substantial computational costs, such as repeated gradient computations or auxiliary model training, which can limit their scalability. We discuss these approaches and the costs they incur in this section.

### F.1 Notations

1. A training dataset $D = \{x_i\}_{i=1}^{N}$ of size $N$; the final language model to be trained on the selected data $\theta$.

2. We denote **the average cost of one forward pass** of model $\theta$ on a training example as $F_\theta$. As one backward pass is approximately the cost of two forward passes, **the average cost of one "gradient pass"** (i.e., one forward + one backward) is thus $3F_\theta$.

3. Another important source of computational cost in data selection comes from the training of additional models. We use $C(\theta, D, T)$ to denote the cost of training model $\theta$ on dataset $D$ for $T$ epochs (i.e., $N \cdot T$ examples are seen in total).

4. Therefore, we unify the computational cost of most data selection approaches into two parts:

   (a) **The training of additional models**. For example, gradient-based influence requires training an additional model on part of the training dataset for $T$ epochs to obtain the checkpoints for gradient computation.

   (b) **The computation of per-sample features**. For example, for each training example, gradient-based influence requires computing its gradient for each saved checkpoint, which means $T$ gradient passes are needed.

5. Note that some algorithms may have additional computational costs other than the two parts above, such as clustering or a greedy algorithm for the final data selection. Since the two parts above constitute the majority of computation for almost all the data selection approaches, **we omit the other cost and only focus on these two.**

## F.2 TLDR: The Final Table

For GRAPE, we assume that in the training dataset $D$, various responses to the same instruction are already available, thus no additional cost is incurred in the *Response Collection* step of GRAPE. So the computational cost analysis of GRAPE under our framework is:

- **Additional Training**: 0, as GRAPE directly evaluates data using the base model.
- **Per-sample conditional probability**: $NF_\theta$, as for a given target model $\theta$, we only need to compute conditional probability for each response (example) once.

The table below shows that our method, GRAPE, achieves superior performance with minimal computational cost compared with other model-based data selection approaches.

| | Additional Training | Per-Sample Feature Computation |
|---|:---:|:---:|
| **GRAPE (ours)** | 0 | $NF_\theta$ |
| Gradient-based influence (LESS) | $C(\theta_{\text{lora}}, D_{\text{warmup}}, T)$ | $3T \cdot NF_\theta$ |
| In-run gradient-based influence | $C(\theta, D, 1)$ | 0 |
| Gradient matching | $C(\theta_{\text{lora}}, D_{\text{warmup}}, T)$ | $3T \cdot NF_\theta$ |
| Gradient norm | $m \cdot C(\theta, D, 1)$ | $3m \cdot NF_\theta$ |
| Embedding-based | 0 | $NF_\theta$ |
| Simple uncertainty indicators | 0 | $NF_\theta$ |
| Perplexity | $C(\theta_{\text{ref}}, D_{\text{ref}}, 1)$ | $NF_{\theta_{\text{ref}}}$ |
| Learnability | $C(\theta, D, 1)$ | $2 \cdot NF_\theta$ |
| Loss trajectory (S2L) | $C(\theta_{\text{ref}}, D, T)$ | $T \cdot NF_{\theta_{\text{ref}}}$ |

Table 13: Computational cost comparison of data selection methods.

## F.3 Gradient-based Methods

Gradients have long been an important source of information for training data selection, as they directly affect the whole optimization process of language models. Three kinds of model-based gradient-based data selection approaches have been proposed:

1. Gradient-based influence
2. Gradient matching
3. Gradient norm

### F.3.1 Gradient-based Influence

Gradient-based influence computes the pairwise influence scores between each pair of training and validation examples. Training data with the highest influence are selected, as training on them leads to the theoretically largest decrease in model loss on validation data. LESS Xia et al. (2024) formulates the pairwise influence scores as the cosine similarity between the gradients of training and validation data, and computes these gradient features using the following two steps:

1. LoRA-train the final model on part of the whole training dataset, denoted as $D_{\text{warmup}}$, for $T$ epochs, and save the $T$ model checkpoints.
2. For each data point, compute its LoRA gradient with each of the $T$ checkpoints, and later aggregate these $T$ gradients together in the cosine similarity expression.

Therefore, the computational cost of gradient-based influence is:

- **Additional training**: $C(\theta_{\text{lora}}, D_{\text{warmup}}, T)$.
- **Per-sample gradient for each checkpoint**: $NT \cdot 3F_\theta = 3T \cdot NF_\theta$.

In order to reduce the cost incurred by per-sample gradient computation, recent work has developed *in-run gradient-based influence* that directly computes the dot product between gradients without

the need for separate gradient computations. However, this approach incorporates the dot product computations into the standard training process, which means in order to obtain pairwise influence scores for **the whole training set**, **a full training run** has to be done on all the training data. This incurs inefficiency when we do not actually need full dataset training. Moreover, the pairwise scores here only show the model's **"dynamic preference"**: scores computed at the $t$-th iteration only reflect the model's preference at this specific iteration. It is not theoretically guaranteed that these scores reflect the model's preference from the beginning of training. Thus, the cost of in-run gradient-based influence is:

- **Additional Training**: $C(\theta, D, 1)$.
- **Per-sample gradient**: 0.

### F.3.2  Gradient Matching

Gradient matching also requires per-sample gradients, but utilizes their information in a different way. It performs clustering based on these gradient features to group similar data, and then applies an iterative greedy selection algorithm. **In order to scale to LLM-level gradient computation and clustering, TAGCOS Zhang et al. (2024b) completely follows the warmup training and gradient computation pipeline of LESS Xia et al. (2024).** As the computational bottleneck here is still the gradient computation instead of clustering or iterative selection, Zhang et al. (2024b) also shares the same computational cost as Xia et al. (2024):

- **Additional training**: $C(\theta_{\text{lora}}, D_{\text{warmup}}, T)$.
- **Per-sample gradient for each checkpoint**: $NT \cdot 3F_\theta = 3T \cdot NF_\theta$.

### F.3.3  Gradient Norm

The $L_2$-norms of gradient vectors can also serve as effective indicators for data selection. Paul et al. (2023) proposes GraNd, which obtains a utility score for each training point based on its gradient norm early in the training. More specifically, it starts from $m$ different model weight initializations, trains each model on the whole dataset to obtain per-sample gradient norms, and finally averages the $m$ gradient norms for each training point to obtain the final GraNd score. Therefore, the computational cost of GraNd is shown below:

- **Additional training**: $m \cdot C(\theta, D, 1)$.
- **Per-sample gradient for each weight initialization**: $Nm \cdot 3F_\theta = 3m \cdot NF_\theta$.

### F.4  Embedding-based Methods

Embedding-based methods project the whole training set into an embedding space to quantify the information of each data point and their interactions. For model-based embedding-based selection methods, the embeddings are usually computed by the final model $\theta$ to align with its preference.

Under a supervised data selection setup where validation data representing target task distributions are available, **Representation-based Data Selection** (RDS; Rubin et al. (2022); Hanawa et al. (2021)) computes the embedding similarity between training and validation data, and selects training points that are most similar to the target distribution in the embedding space.

For an unsupervised setup where only the embeddings of training data are accessible, **geometry-based coreset sampling** methods are widely used Qin et al. (2024). Grounded on the intuition that close samples in the embedding space often share similar properties, a **diverse** subset can be obtained by controlling the minimum distance between any two selected data points. Among them, using K-center greedy sampling to select embedding-based **facility locations** has been proven especially effective for instruction fine-tuning of LLMs Bhatt et al. (2024).

These embedding-based approaches share similar computational costs: they do not need any additional model training and can directly extract useful per-sample embeddings using the last-layer hidden states of the pretrained final model $\theta$. Thus, their computational cost is shown below:

- **Additional training**: 0.
- **Per-sample embedding computation**: $NF_\theta$.

## F.5 LogProb-based Methods

LogProb-based methods also directly utilize the target LLM to evaluate the utility of each training data point.

### 4.1 Simple Uncertainty-based Indicators

Some simple model-based indicators inspired by the notion of uncertainty have been shown effective for a long time and recently extended to data selection for LLM instruction tuning (Marion et al., 2023b; Bhatt et al., 2024). Bhatt et al. (2024) demonstrates the effectiveness of various indicators including **mean entropy, least confidence, mean margin**, etc. These simple indicators do not require additional training and can also be directly obtained with the pretrained final model $\theta$. Their computational cost is shown below:

- **Additional training**: 0.
- **Per-sample per-token logits computation**: $NF_\theta$.

### F.5.1 Perplexity (PPL)

PPL is also a long-standing data selector and has been shown effective for LLM-scale data selection. Typically, a split of the training dataset, $D_{\text{ref}}$, is needed to train $\theta_{\text{ref}}$, a reference model that will be used to compute PPL for the whole training set.

A common approach is to use the final model $\theta$ as the reference model $\theta_{\text{ref}}$ to ensure the alignment in PPL patterns Marion et al. (2023a), but prior work Ankner et al. (2024) also shows that a reference model much smaller than the final model can also be an effective PPL-based data selector. The computational cost for PPL-based selection is shown below:

- **Additional training**: $C(\theta_{\text{ref}}, D_{\text{ref}}, 1)$.
- **Per-sample PPL computation**: $NF_{\theta_{\text{ref}}}$.

### F.5.2 Learnability

In addition, **learnability (Mindermann et al., 2022; Zhou et al., 2024a,b)** is a more effective metric than pure uncertainty or PPL, as it excludes uncertain but unlearnable points (e.g., noisy or less task-relevant) by considering the decrease in per-sample loss before and after the model is fully trained. More specifically, it trains the final model $\theta$ on the full training dataset to obtain a strong reference model $\theta_{\text{ref}}$, and then computes the difference of loss on each training example between $\theta$ and $\theta_{\text{ref}}$. In this way, it requires two forward passes for per-sample computation:

- **Additional training**: $C(\theta, D, 1)$.
- **Per-sample learnability computation**: $2 \cdot NF_\theta$.

### F.5.3 Loss Trajectory

Moreover, logprob-based methods can also obtain finer-grained information from the **training dynamics** of LLMs. S2L Yang et al. (2024b) obtains a feature vector for **each** training point by collecting their **training loss trajectories** over $T$-epoch training on a small reference model $\theta_{\text{ref}}$, and then applies K-means clustering to equally sample data points from each trajectory cluster. Prior work shows its superiority over other logprob-based indicators, but it also comes with significant computational cost:

- **Additional training**: $C(\theta_{\text{ref}}, D, T)$. Here the choice of $\theta_{\text{ref}}$ is especially important, as prior work Xia et al. (2023) shows that reference models that come from the same model family as the final model tend to have similar loss trajectories of training data, so they can preserve more fidelity in their loss trajectory patterns.
- **Per-sample loss trajectory computation**: $T \cdot NF_{\theta_{\text{ref}}}$. Note that $T$ here is typically much larger than that in gradient-based influence computation, so the computational cost of this gradient-free approach can be even higher than gradient-based methods.

## G    Multi-Round GRAPE

To evaluate whether GRAPE continues to provide benefits after an initial round of fine-tuning, we conducted a second-round experiment using LLaMA3.1-8B. In the first round, we selected the best responses per instruction using GRAPE and fine-tuned the model accordingly. Then, we re-applied GRAPE on the newly fine-tuned model to select a fresh set of responses and conducted another round of fine-tuning. This second iteration led to further performance improvements across multiple benchmarks, including AlpacaEval, WizardEval (both original and LeetCode), and real-world tasks such as MATH, MMLU, and BBH. Notably, GRAPE Round 2 improved average benchmark scores from 35.9% to 37.3%, demonstrating that the method remains effective and even compounding when iteratively applied.

| Method | AE | WR | WR (LC) | MATH | MMLU | BBH |
|---|---|---|---|---|---|---|
| GRAPE Round 1 | 15.2 | 14.8 | 19.4 | 32.1 | 64.5 | 69.6 |
| GRAPE Round 2 | **17.6** | **19.6** | 18.9 | **33.2** | 64.5 | **70.0** |

Table 14: Performance of LLaMA3.1-8B after two rounds of GRAPE fine-tuning.

## H    Details Of Correlation Analysis

To analyze how well different models align with the training distribution, we conducted a perplexity-based study on the Tulu-v3 training set. Specifically, we randomly sampled 1,000 instances from the training data and used a wide array of generator models to produce responses for each instruction. For each response, we computed its perplexity under each base model and ranked the responses by perplexity, with lower perplexity indicating better matching of distribution. We then identified the top-1 response per instance for each base model and visualized the overall rankings using a heatmap (Figure 4).

