# OpenReview forum: "The Best Instruction-Tuning Data are Those That Fit"
_NeurIPS.cc/2025/Conference — NeurIPS 2025 spotlight_

### Official Review · Reviewer_qwim · 2025-06-10

**Clarity:** 4
**Significance:** 3
**Originality:** 4
**Rating:** 5
**Confidence:** 5

**Summary:**

The paper introduces a simple strategy to select which response to SFT with -- ones that are the most aligned with the base, pretrained model (average conditional probability). Experiments explore diverse and realistic settings where such selection can be applied and demonstrated superior performance in all the settings. The compared baselines cover a wide range of common response selection strategies.

**Questions:**

Please see the two points raised in ``Weaknesses'' above.

**Ethical Concerns:**

["NO or VERY MINOR ethics concerns only"]

**Final Justification:**

The authors issued some clarification questions during rebuttal but wasn't able to provide a satisfactory answer to the limitations on the types of model that their method will remain effective. Specifically, when a pretrained model is poorly trained and assigns high likelihood to truly inferior responses, trading off the quality and learnability is relevant. Thus, my rating remains the same.

**Limitations:**

Not quite. See point 2 of ``Weaknesses''.

**Quality:**

4

**Strengths And Weaknesses:**

Strengths:
* Clarity: The paper is well written and easy to follow. Hypotheses are explicitly and verified. The proposed method is simple and straightforward.
* Quality: Experiment settings and baselines are extensive. The appendix covers extended ablation studies of their proposed technique.
* Originality: The selection of responses for instruction-tuning have not been formally studied to the extend of my knowledge.
* Significance: The topic is relevant and the experiment results show that the proposed technique is consistently effective.
* Rigor in implementing baselines: Details of properly implementing baselines (mentioned in Appendix C) is much appreciated.

Weaknesses:
* Intuition: selecting the responses that are most aligned with the base pretrained model over the ``best'' responses (e.g. generated from the strongest model or ranked by external reward model) is counterintuitive. Providing some more intuition, perhaps in the form of simple theoretical analysis explaining the mechanism would yield more insight.
* Missing Limitations section: The checklist suggests the limitations are provided in Sec 5.6 but said section does not exist. It is perhaps unlikely that GRAPE will work with *all* pretrained models. For instance, smaller language models (1B, 3B) may not have sufficient internal knowledge to provide effective signal for response selection. Analyzing the limitations of the applicability of the proposed method is relevant for real-world usage.

---

> ### Author Rebuttal · Authors · 2025-07-31
>
> Thank you so much for your thoughtful and generous review!  We appreciate your recognition of the strength of our approach.
>
> ---
> ### Intuition Behind GRAPE
>
> GRAPE is motivated by the direct analogy to the benefits of on-policy reinforcement tuning  and preference learning (for example,[1,2])  where it is well understood that on-policy updates can well preserve the model’s pre-trained capabilities and improve learning stability and generalization[3]. Recent studies point out that these on-policy updates can achieve the effect of updating only parameter sub-groups of the pre-trained model and thus preserving its pre-trained knowledge while acquiring new capabilities [4]. In parallel, recent supervised fine-tuning [5] work shows that not all high-quality responses are equally “learnable” for a given base model: stronger teacher models do not always provide the best supervision for downstream fine-tuning , and small models in particular can struggle when learning from overly complex reasoning chains produced by larger reasoners.
> GRAPE builds directly on these intuitions. It aims to identify responses that are well-aligned with the target model’s own distribution, ensuring that supervision during SFT is both accessible and effective. We provide a detailed discussion of this motivation in Section 2.
>
> While our focus here is empirical, we agree that developing a deeper theoretical understanding of how on-policy alignment benefits LLMs across post-training stages is a fascinating and valuable direction for future work!
>
> ---
> ### Model Sizes
>
>
>
> We appreciate your thoughtful consideration about GRAPE’s applicability boundary. We did experiment with representative models of varying sizes - Qwen-1.5B, Qwen-3B, and LLaMA3.2-3B (Table 1 in the main text; Appendix Tables 9–10) show that GRAPE continues to offer strong performance on these smaller models, which adds encouraging evidence of GRAPE’s generality.
>
> That said, the reviewer's consideration is definitely valid and we will clearly note this point in the limitation section!
>
> ---
> ### Limitations
>
>
> Thank you for catching this—we apologize for the oversight. Due to space constraints, our limitations discussion was truncated waived into a paragraph under Section 5.5 (“Why GRAPE Outperforms Self-Generated Responses”), where we caution against over-relying on self-generated in-distribution data. We will add a standalone Limitations section in the revised version to thoroughly discuss the limitations and incorporate the feedback (for example, the model-size spectrum)!
>
> ---
> Once again, we sincerely appreciate your constructive and insightful feedback!
>
> ---
>
>
> [1] Tajwar et al. 2024. Preference Fine-Tuning of LLMs Should Leverage Suboptimal, On-Policy Data. https://arxiv.org/abs/2404.14367.
>
> [2] Xiong et al. 2025.  A minimalist approach to llm reasoning: from rejection sampling to reinforce. https://arxiv.org/pdf/2504.11343.
>
> [3] Meng 2025. Good Actions Succeed, Bad Actions Generalize: A Case Study on Why RL Generalizes Better. https://arxiv.org/pdf/2503.15693.
>
> [4] Mukherjee et al. 2025. Reinforcement Learning Finetunes Small Subnetworks in Large Language Models. https://arxiv.org/pdf/2505.11711.
>
> [5] Li et al. 2025. Small Models Struggle to Learn from Strong Reasoners. https://arxiv.org/pdf/2502.12143

---

> > ### Comment · Reviewer_qwim · 2025-08-06
> >
> > Thank you for the rebuttal. Regarding the motivation, it is still odd that purely optimizing for the "learnability" is sufficient. Consider a scenario where the pretrained model is poorly trained and assigns high likelihood to truly inferior responses. In this case, trading off the quality and learnability is relevant. This echoes Reviewer WaRw's point that the models experimented are all decently good to begin with, which allows such one-sided greedy heuristics to succeed. This should be included into the limitations section.

---

### Official Review · Reviewer_YZYx · 2025-07-03

**Clarity:** 4
**Significance:** 4
**Originality:** 4
**Rating:** 5
**Confidence:** 4

**Summary:**

This papers propose GRAPE, and efficient and simple framework for instruction tuning data selection. The authors draws the insights from reinforcement learning to hypothesize that training model with response distrubution closer to pre-training can leads to better perofrmance. They further propose GRAPE, a two step framework that collect instruction-responses from multiple-sources and then select the responses that is most similar to the training models' distrubition by measuring the perplexity. The comprehensive evaluation shows significant performance improvement on a wide range of models and benchmarks.

**Questions:**

1. I have a question about the hypothesis that proposed in this paper, which by selecting response that is more similar to the pre-training distribution can help performance. In some of the prior work such as [1], they show that selecting instructions that are more out-of-distribution can leads to better cross-task generalization. From both line of the findings, do you think that better instruction data should have more o.o.d instructions, with responses that is more similar to the training model?
2. Following 1, if training on o.o.d tasks, do you think that it will distort the pre-training distribution more and leads to lower performance following the hypothesis proposed in this paper?



[1] Active Instruction Tuning: Improving Cross-Task Generalization by Training on Prompt Sensitive Tasks, EMNLP2023

**Ethical Concerns:**

["NO or VERY MINOR ethics concerns only"]

**Final Justification:**

This paper discuss how response distrubution closer to pre-training models can leads to a better perofrmance. The perspective is novel and can be useful in realistic scenarios for instruction tuning data selection and generation, which I found very interesting. I will remain my original score.

**Limitations:**

yes

**Quality:**

4

**Strengths And Weaknesses:**

# Stength
1. The paper is well written and easy to follow.
2. The hypothesis of SFT data that is closer to pre-training distrubtion can be more effective is insightful.
3. The proposed method is extremely simple and can generalize across scenarios. The comprehensive evaluation shows that GRAPE framework can boost performance on a wide range of models and benchmarks.
4. The insight that SFT data quality can largely depends on the training model, while have been discusses in some prior work in other perspectives (such as selecting OOD data can leads to better cross-task generalization), is very important for the synthetic data generation/selection field. The inisight of training on data that is more similar to pre-training distribution is novel and supported by empirical results, can potentially brings huge impact to this research direction.

# Weakness
No obvious weakness.

---

> ### Author Rebuttal · Authors · 2025-07-31
>
> We sincerely thank the reviewer for the thoughtful and generous feedback!
>
> > Q1: From both lines of the findings, do you think that better instruction data should have more o.o.d instructions, with responses that are more similar to the training model?
>
> Thanks for discussing the outlook for a composite strategy with us!
>
> We believe these two lines of findings could be complimentary. Active Instruction Tuning (AIT) [1] focuses on task selection, identifying “ambiguous” instructions (with high prompt sensitivity to the model) that support better cross-task generalization. GRAPE, on the other hand, focuses on response selection—choosing responses that align well with the base model’s own distribution to facilitate stable and effective fine-tuning. Thus, we envision that these two approaches can yield synergistic effects and we are happy to include discussion on AIT in our next revision!
>
> > Q2: Following 1, if training on o.o.d tasks, do you think that it will distort the pre-training distribution more and leads to lower performance following the hypothesis proposed in this paper?
>
> Thanks for the insightful follow-up discussion on compatibility of these task selection and response selection strategies.
>
> We do not see AIT’s approach conflicting with GRAPE’s intuition. AIT shows that ambiguous tasks (high prompt sensitivity) improve generalization, while difficult tasks (high aggregated perplexity, low sensitivity) do not have such benefits and could even harm (see, for example, Figure 7 in their paper). The latter half in fact echoes GRAPE’s hypothesis. Thus, we conjecture that applying their task selection criterion does not necessarily distort the model distribution more. Given that GRAPE is effective across diverse instruction datasets (e.g., Tulu-OLMo, UltraInteract, OpenR1, OpenHermes-2.5, Magpie), we believe a combined strategy could harness the strengths of both approaches.
>
> ---
>
> Once again, we sincerely thank the reviewer for the thoughtful and encouraging review. Your insights have helped us better position our contributions and think more deeply about the broader implications.
>
> [1] Active Instruction Tuning: Improving Cross-Task Generalization by Training on Prompt Sensitive Tasks, EMNLP2023

---

### Official Review · Reviewer_CqUm · 2025-07-03

**Clarity:** 4
**Significance:** 3
**Originality:** 3
**Rating:** 5
**Confidence:** 4

**Summary:**

The paper proposes an improved method for supervised finetuning in the presence of multiple possible labels (e.g. in the distillation setting). In this setting, the paper proposes ranking possible ground-truth responses by their distributional alignment with the base model. The authors nicely draw an analogy to on-policy reinforcement learning to justify this. This setting of having access to multiple labels for each instruction emerges naturally during model distillation - for distillation. The proposed method (“GRAPE”) generally outperforms distillation from the massive Llama-3.1-405B model, as well as all other baselines.

**Questions:**

- How does this method relate to other on-policy distillation methods, such as self-play finetuning? (Chen, Deng, Yuan et al 2024).
- What is the significance of the "3x Data" baseline in section 4.1? It seems like this is just "Original-UI", but the effective learning rate is increased by 3x.
- Do you expect that performance would be different if you selected the response to use for SFT via a different method, such as minimum Bayes risk decoding (https://arxiv.org/abs/2410.02902)?

**Ethical Concerns:**

["NO or VERY MINOR ethics concerns only"]

**Final Justification:**

Score: 5. Clear accept.

While not "technically flawless", this paper has few flaws, and it's conceptually very interesting. In private conversations with people, I've found myself wanting to tell people about this paper, and it needs to be published to allow that.

Pros:
- Intuitive idea
- Excellent empirical support for the idea

Weaknesses:
- Insufficient baselines, though the baselines given in the current draft are enough to convince me that this method is interesting.

I hope the authors add more baselines in the camera ready, but I think this is a clear accept.

**Limitations:**

There is insufficient discussion of limitations. *The authors' paper checklist states that, for information on Limitations, "see section 5.6, where we show pursuing in-distribution answers in the wrong way can lead to performance degradations.". Section 5.6 does not actually exist.

**Quality:**

4

**Strengths And Weaknesses:**

Strengths:
- The method is empirically very strong.
- The paper is exhaustive in its selection and usage of baselines, including very strong and relatively recent baselines (e.g. Tulu 3, LESS). The proposed method, GRAPE, outperforms these baselines on almost every benchmark.
Weaknesses:
- The most common scenario where multiple labels are available is distillation. This paper only includes one experiment explicitly studying the role of GRAPE when samples can be collected by only a single teacher. I think this ought to be expanded, with additional baselines, such as “always choose the [longest/shortest] response”, or “choose the response with the greatest embedding similarity with the others” (minimum Bayes risk decoding).

---

> ### Author Rebuttal · Authors · 2025-07-31
>
> Thank you for this thorough review! We sincerely appreciate your positive assessment of our empirical contributions and your recognition of the careful baseline comparisons we've included. Your thoughtful questions and suggestions are invaluable in helping us strengthen and clarify our work.
>
> > SFT v.s. Single-Teacher Distillation Set-Up
>
> Thank you for correctly pointing out that our work is primarily under the broader general instruction-tuning (SFT) setting, where candidate responses often come from a diverse pool of sources—ranging from human annotations to mixtures of model-generated outputs.
>
> We agree that the single-teacher distillation setting as a special case and validated the effectiveness in Section 5.4 using Magpie, and R1-style long CoT distillation in Appendix B.3, demonstrating that GRAPE continues to be effective with one single data source. That said, we acknowledge that most of our experiments are conducted in the more general SFT scenario, and we will be sure to clarify this distinction and its implications in the revision!
>
> > Suggested Baselines
>
> Thanks for suggesting the simple heuristic-based baselines! While we are running these experiments but unsure if these results will be available before rebuttal deadline due to limited compute, we did compare GRAPE against a wide array of strong and widely adopted baselines—including reward-based selection, and recent model-dependent strategies like LESS, S2L, and embedding-based strategies, in our submission.
>
> > How does this method relate to other on-policy distillation methods, such as self-play fine tuning?
>
> There is indeed a deep conceptual resonance between our GRAPE framework and SPIN - leveraging the model’s own inductive biases to guide training to reduce catastrophic distribution shift.
>
> Their goals of optimization, however, are different. SPIN pushes the model toward the human data distribution through adversarial training, while GRAPE preserves the model's pretrained distribution by selecting training data that already aligns with it. They fit in different stages of post-training, where GRAPE targets SFT stage and SPIN further improves SFT’ed checkpoint via self-play.
>
> > Do you expect that performance would be different if you selected the response to use for SFT via a different method, such as minimum Bayes risk decoding
>
> Yes, we expect that using Minimum Bayes Risk (MBR) to select SFT responses could lead to different and potentially improved outcomes when the utility function is well-aligned with the downstream evaluation metric as MBR explicitly optimizes for expected performance under a given utility.
>
> We find MBR-based selection relevant and will include a discussion of it in our revision. Exploring hybrids that combine GRAPE’s distributional alignment with task-aware MBR objectives could also be a promising future direction!
>
> > What is the significance of the "3x Data" baseline in section 4.1?
>
> The “3× Data” baseline increases the number of (validated) unique responses per instruction to 3× the original size of UltraInteract, without altering the learning rate or number of epochs. We will clarify this in the revision!
>
> > There is insufficient discussion of limitations.
>
> We apologize for any confusion - due to space constraint, we waived the discussion into the second paragraph titled “Why GRAPE Outperforms Self-Generated Responses” in section 5.5 last-minute, where we pointed out in-distributionness alone (which to its extreme becomes training on base model’s self-generated responses) is insufficient during SFT stage.
> We will add a standalone Limitation and Future Work section to include more detailed discussions on the limitations of our work and incorporate the feedback from the reviewers.
>
> Finally, we would like to thank the reviewer again for their encouraging feedback and in-depth discussions!

---

> > ### Comment · Reviewer_CqUm · 2025-08-06
> > **Thank you for the detailed rebuttal**
> >
> > Thank you for your well-written and detailed rebuttal!
> >
> > One quick clarification:
> >
> > > Thanks for suggesting the simple heuristic-based baselines! While we are running these experiments but unsure if these results will be available before rebuttal deadline due to limited compute, we did compare GRAPE against a wide array of strong and widely adopted baselines—including reward-based selection, and recent model-dependent strategies like LESS, S2L, and embedding-based strategies, in our submission.
> >
> > Where is reward-based selection discussed in the current draft? I tried looking for this and I couldn't find it. I do appreciate the inclusion of data filtering baselines (LESS, S2L), but I think these are orthogonal to the question of how to handle multiple responses.
> >
> > Ultimately, I think this is a nice paper and I enjoyed reading it. My only concern continues to be the lack of some important baselines. I should clarify the reason I would like more baselines. I don't want them for the purposes of demonstrating that the proposed method is "SOTA" - I already am convinced that the proposed method is both effective and intellectually interesting. I want more baselines in order to be able to consider how using the student model's likelihood of a response contrasts with other related techniques (e.g. MBR, SPIN, heuristics for choosing a response). I would really like to see these in a camera ready!
> >
> > But even in their absence, I like this paper, and I will maintain my current strong score of 5.

---

> ### Author Response · Authors · 2025-08-08
>
> We sincerely thank the reviewer for the strong score and the thoughtful, constructive feedback! We greatly appreciate the suggestion to add more baselines for better understanding GRAPE’s relation to methods like MBR, SPIN, and simple heuristics.
>
> Thanks for the clarification on the reward-based baselines!  Reward-based selection results are in Appendix B-1! On both Llama 3.1-8B and Mistral-v0.3-7B, GRAPE outperforms reward-based selection in terms of overall scores.
>
> We agree that adding further baselines would enrich the discussion and will make every effort to include them in the camera-ready, resource permitting.
>
> Thank you again for your encouraging assessment and for helping strengthen this work!

---

### Official Review · Reviewer_WaRw · 2025-07-05

**Clarity:** 3
**Significance:** 4
**Originality:** 3
**Rating:** 5
**Confidence:** 4

**Summary:**

The paper “The Best Instruction-Tuning Data are Those That Fit” investigates how to select better data for instruction fine-tuning large language models (LLMs). The authors argue that high-quality responses generated by external, stronger models are not necessarily ideal for training a weaker target model, because these responses can be out of distribution and harder for the target to learn. They propose GRAPE, a method that collects multiple candidate responses for each instruction and then uses the target model itself to evaluate and select the response it finds most likely, effectively choosing data that fits its own distribution. Experiments show that this approach improves both performance and robustness of the fine-tuned model, demonstrating that aligning training data with the target model’s prior knowledge is more effective than simply choosing the most sophisticated external responses. The paper challenges the assumption that better external labels always lead to better fine-tuning outcomes.

**Questions:**

I am wondering if the proposed method will still work if the target LLM’s base performance is not ideal. The three models, LLaMA 3.1‑8B, Mistral‑7B, and Qwen2.5‑7B, are already relatively strong open models, so it remains unclear how effective GRAPE would be when applied to much weaker or less capable target models that may struggle even with well-aligned data. It would also be valuable to analyze the behaviors of these three models, as understanding how model-specific factors influence GRAPE’s effectiveness could provide deeper insight.

**Ethical Concerns:**

["NO or VERY MINOR ethics concerns only"]

**Final Justification:**

I will keep my current score.

**Limitations:**

A limitation of this paper is that its experiments and conclusions are confined to the chain-of-thought reasoning task, which raises questions about how well the proposed approach generalizes to other instruction-tuning settings or task types beyond CoT.

**Paper Formatting Concerns:**

The borders around the clickable cross-reference/linked items need to be removed.

**Quality:**

3

**Strengths And Weaknesses:**

One key strength of the paper is its clear focus on an important yet often overlooked aspect of instruction fine-tuning: selecting data that is well-suited to the target model rather than simply relying on higher-quality external responses. This highlights the critical role of data–model alignment in effective instruction tuning. Another strength is the thoroughness of the experimental evaluation; the authors compare against strong and diverse baselines, including controlled, scaling and other data selection approaches which lends credibility and robustness to their conclusions about the benefits of their proposed approach.
A limitation of this paper is that its experiments and conclusions are confined to the chain-of-thought reasoning task, which raises questions about how well the proposed approach generalizes to other instruction-tuning settings or task types beyond CoT.

---

> ### Author Rebuttal · Authors · 2025-07-31
>
> We thank the reviewer for their positive and thoughtful feedback and your recognition of our focus on data–model alignment and the breadth of our evaluations. We seek to answer the questions below –
>
> > How effective GRAPE would be when applied to much weaker or less capable target models that may struggle even with well-aligned data
>
> We agree that this is an important piece. While most results from our main tables focus on 7B-scale models, we also included smaller models like LLaMA3.2-3B and Qwen1.5B/3B (Table 1 and Table 9). Encouragingly, we observe consistent improvements even at these smaller scales—for example, GRAPE improves LLaMA3.2-3B by 4% on UltraInteract, providing positive evidence that GRAPE works for varying model sizes. That said, we acknowledge that the model range we experimented with is still constrained, and we will explicitly acknowledge this potentially limited model-size spectrum in our limitations section!
>
> > Understanding model-specific behavior
>
> Thank you for seeking more discussion on model-specific behavior. Figure 5 presents a correlation analysis across models, showing that different base models indeed will prefer different responses for the same instruction, and a consistent trend that each model is the most likely to select the response produced by its instruction-tuned counterpart compared to other models (e.g. Qwen2.5-7B-Base picks its instruction-tuned counterpart in 39.7% of all instructions, and this number varies across models)., as discussed in Appendix H. This could partially explain the finding in [1] that models benefit from response data from the same model family.
>
> > how well the proposed approach generalizes to other instruction-tuning settings or task types beyond CoT.
>
> Thanks for the opportunity for us to clarify the experimental set-up! We indeed tested GRAPE on more general instruction-tuning settings.
>
> While Section 4 focuses on CoT for controlled analysis, Section 5 evaluates GRAPE in broader, real-world instruction-tuning settings. We apply it to large-scale general-purpose instruction-tuning datasets like the Tulu-3/OLMo-2 mixture (Section 5) and OpenHermes-2.5 (Appendix B.2), which span diverse domains including open-ended dialogue, QA, coding, safety, and general instruction following -- no longer confined to CoT style.
>
> On the evaluation side, in Section 5 we incorporated both CoT-reasoning like coding and math reasoning; and non-CoT-reasoning benchmarks like MMLU (knowledge and multitask understanding) and AlpacaEval2 (human alignment), shown in Tables 2 and 3, which GRAPE consistently yields strong performance on.
>
> We will clarify the dataset compositions and evaluation coverage more clearly in the revision!
>
> Once again, thank you for your constructive, encouraging review. We’re excited to refine our manuscript and look forward to incorporating your feedback in the final version!
>
> ---
> [1] Xu et al. 2024. Stronger Models are NOT Stronger Teachers for Instruction Tuning. https://arxiv.org/abs/2411.07133

---

> > ### Comment · Reviewer_WaRw · 2025-08-05
> >
> > Thanks to authors for the detailed response. It addressed my questions and concerns. I will keep my current score.

---

> > > ### Author Response · Authors · 2025-08-08
> > >
> > > Thank you for your positive and encouraging assessment of our work!

---

### Note · Authors · 2025-08-16

We thank the reviewers for their thoughtful engagement and consistently strong assessments! We’re encouraged by the shared recognition that our work offers a principled and practical answer to an important SFT question: which response a model should learn from.

## Summary of Acknowledged Strengths

- **Core Insight & Novelty**
Reviewers appreciated our key idea: learning from in-distribution responses improves instruction tuning. It’s seen as intuitive, original, and impactful.

- **Simplicity & Practicality**
GRAPE is lightweight, training-free, and integrates seamlessly into standard SFT pipelines.

- **Strong Empirical Results**
GRAPE consistently outperforms reward-based selection, strong-teacher distillation, and heuristic methods across diverse models (e.g., LLaMA-3, Mistral, Qwen) and datasets (e.g., Tulu, UltraInteract).

- **Thorough Evaluation**
Reviewers highlighted our extensive comparisons (e.g., LESS, S2L), detailed ablations, and evaluations across model families and sizes.

- **Clarity & Presentation**
The paper was noted for its clear hypotheses, structured narrative, and accessible writing.



## Concerns Addressed in Rebuttal

- **Relation to Prior Work**
We clarified GRAPE’s relation to AIT, SPIN, and MBR, and will expand this discussion in the revision.

- **Motivation**
GRAPE is motivated by on-policy learning principles: selecting in-distribution responses reduces distribution shift and enhances generalization by preserving pre-training behavior.


## Planned Improvements for the Next Revision

### Limitations Section
We will include a standalone limitation section and explicitly state potential considerations with GRAPE:
- It assumes a quality-controlled candidate pool.
- The selection effectiveness might be influenced by base model capabilities.

### Expanded Related Work
- Discuss SPIN, AIT, and MBR.
- Clarify complementarity: GRAPE (SFT data curation), SPIN (self-play policy optimization post-SFT).
- Differentiate GRAPE (response selection) from AIT (instruction selection), and discuss potential synergies.
- Discuss the difference between GRAPE and utility-driven methods like MBR.

### Presentation & Clarity Enhancements
- Clarify "3× Data" setup.
- Emphasize coverage of both Chain-of-Thought (Sec. 4) and general instruction tuning (Sec. 5).
- Reinforce the assumption of a quality-controlled response pool.

We will disclose documentation and code to support replication and follow-up research. Thanks all the reviewers again!

---

### Decision · Program_Chairs · 2025-09-17

**Decision:**

Accept (spotlight)

**Comment:**

The paper introduces a simple strategy, named GRAPE, to select which response to use for supervised fine-tuning (SFT) of LLMs. In particular, when multiple responses to the same instruction are available, e.g. generated from multiple sources, GRAPE selects the most aligned one with the base, pre-trained model, i.e. the one with lowest perplexity according to the base model. The extensive experiments explore diverse and realistic settings where such selection can be applied, and include multiple datasets, tasks and models. The proposed approach demonstrates superior performance across settings compared to various baselines covering a wide range of common response selection strategies.

All reviewers found the proposed method simple, novel and insightful, especially with regard to selecting the SFT data depending on their alignment to the base model. Moreover, the empirical results are strong, and the paper well-presented.

During the rebuttal period, the authors clarified differences to previous works, missing baselines, and the experimental setup, addressing most concerns of the reviewers, and promised to integrate such improvements into the manuscript. As unsolved points, **Reviewer WaRw** and **Reviewer qwim** question the effectiveness of GRAPE on lower-quality base models, and **Reviewer CqUm** would like to see more baselines included.

Overall, all reviewers are very positive about the paper, both in terms of method and evaluation. The simplicity and effectiveness of GRAPE can be practically impactful, especially if, as argued by the authors, can be combined with other data selection methods. Moreover, the idea behind it may be the basis for future work on data selection for supervised fine-tuning. The remaining weaknesses seem relatively, and can be addressed in the final version. Therefore, the paper can be very interesting for the community.